# Relationship between Residual Urine Output and Type of Dialysis with FGF23 Levels

**DOI:** 10.3390/jcm12010222

**Published:** 2022-12-28

**Authors:** Valentina Corradi, Sara Samoni, Alice Mariotto, Carlotta Caprara, Elisa Scalzotto, Anna Chiara Frigo, Francesca K. Martino, Davide Giavarina, Claudio Ronco, Monica Zanella

**Affiliations:** 1Department of Nephrology, Dialysis and Transplantation, San Bortolo Hospital, AULSS 8 BERICA Vicenza, 36100 Vicenza, Italy; 2International Renal Research Institute of Vicenza and IRRIV Foundation for Research, San Bortolo Hospital, AULSS 8 BERICA Vicenza, 36100 Vicenza, Italy; 3Department of Nephrology and Dialysis, S. Anna Hospital, ASST Lariana, 22077 Como, Italy; 4Department of Cardiac, Thoracic, Vascular Sciences and Public Health, University of Padova, 35122 Padua, Italy; 5Department of Laboratory Medicine, San Bortolo Hospital, AULSS 8 BERICA Vicenza, 36100 Vicenza, Italy; 6University of Padova, 35122 Padua, Italy

**Keywords:** hyperphosphatemia, fibroblast growth factor 23 (FGF23), residual urine output (UO), chronic kidney disease (CKD), hemodialysis (HD), peritoneal dialysis (PD)

## Abstract

Several studies investigated the role of fibroblast growth factor 23 (FGF23) in the regulation of renal phosphate excretion in chronic kidney disease (CKD). However, patients with residual urine output (UO) seem to control their serum phosphorus levels better. Our aim was to determine whether FGF23 levels are influenced by dialysis modality and UO. We performed a cross-sectional study in hemodialysis (HD) and peritoneal dialysis (PD) patients. The C-terminal FGF23 (cFGF23) levels were determined in plasma with a two-site enzyme-linked immunosorbent assay. The UO collection referred to an mL/day measurement. All *p* values were two-sided, and the statistical significance was set at *p* < 0.05. We enrolled 133 patients (58 HD, 75 PD, UO 70%). The median cFGF23 was significantly higher in HD vs. PD patients (*p* = 0.0017) and not significantly higher in patients without UO (*p* = 0.12). We found a negative correlation between cFGF23 and the UO volume (*p* = 0.0250), but the correlation was not significant when considering the type of dialysis treatment. Phosphorus (ß = 0.21677; *p* = 0.0007), type of dialysis (ß = −0.68392; *p* = 0.0003), and creatinine (ß = 0.08130; *p* = 0.0133) were significant and independent predictors of cFGF23 levels. In conclusion, cFGF23 was significantly higher in HD than in PD patients. We found a significant negative correlation between cFGF23 and the residual UO volume, but the correlation was not significant considering the type of dialysis. Our study reveals that dialysis modality is an independent predictor of FGF23 levels. In particular, PD is associated with lower FGF23 levels than HD.

## 1. Introduction

Mineral and bone disorders (MBDs) are prevalent in chronic kidney disease (CKD), especially in the advanced stages, representing a severe complication and an important mortality risk factor in both hemodialysis (HD) and peritoneal dialysis (PD) patients [1].

The discovery of the “fibroblast growth factor 23 (FGF23)-bone-kidney axis” has led to several investigations to better understand this complex endocrine network which plays a crucial role in MBD development [2].

FGF23 is a 32 kDa glycoprotein secreted by osteocytes in response to increased levels of 1,25 dihydroxy vitamin D (1,25(OH)2D) [3], while the stimulating roles of phosphate load [4,5] and the parathyroid hormone (PTH) [6] are still debated. The FGF23 intact protein, in the presence of its coreceptor klotho, acts as a counter-regulatory hormone to 1,25(OH)2D [7] and inhibits phosphate reabsorption in renal tubules [8], acting synergistically with PTH.

Despite the fact that no phosphate sensor has yet been identified, most authors agreed that, in CKD, the reduced nephron mass causes a decreased renal phosphate excretion that leads to increased FGF23 secretion from the bone to compensate for hyperphosphatemia [9]. It was demonstrated that FGF23 rises early in CKD, in advance of other alterations [9,10], and in acute kidney injury (AKI) [11,12].

The growing attention on FGF23 is partly due to the finding of its correlation with cardiovascular mortality and CKD progression. Concerning cardiovascular risk, a significant correlation was found between the FGF23 level and arterial calcification and stiffness [13], valve calcification [14], and left ventricular hypertrophy [15]. Regarding the second point, FGF23 was found to be the only independent risk factor for CKD progression among CKD-MBD biomarkers [16].

Some studies have shown that both HD and PD with residual urine output (UO) have better maintenance of phosphate balance than those without UO [17,18,19]. Moreover, a few recent studies have suggested a negative correlation between residual UO and FGF23 levels [18,19,20].

On the other hand, there is no consensus about the phosphate balance in PD and HD patients, as some studies have shown better phosphate control in PD patients [17,21], while other studies observed higher phosphatemia in PD than in HD patients [22]. Hyperphosphatemia is frequently found in patients with CKD, as the progressive reduction in nephrogenic mass leads to a significant marked decrease in the ability of renal phosphorus excretion. The analysis of Bi and colleagues [23] demonstrated that FGF23 levels are significantly higher in HD than in PD patients.

The purpose of the present study was to examine the influence of dialysis type and the presence of residual UO on FGF23 levels.

## 2. Materials and Methods

### 2.1. Study Population

We performed a cross sectional-study in two groups of patients: HD and PD patients. We enrolled patients that underwent HD or PD treatment for at least 3 months with age ≥ 18 years and who agreed to participate in the study between October 2014 and October 2015.

Inclusion criteria

Age ≥ 18 years;

Patients in chronic dialysis HD or PD (≥3 months);

Informed consent was taken before enrollment.

Exclusion criteria

Dementia;

Patients with missing or lacking anamnesis;

Hepatic encephalopathy;

Heart failure New York Heart Association (NYHA) functional classification IV;

Acute myocardial infarction (IMA);

Acute coronary syndrome (ACS);

Lower respiratory tract infection (LRTI);

Systemic inflammatory response syndrome (SIRS);

Received immunosuppressive agents;

Pregnant.

This study was performed with the ethical principles of the Declaration of Helsinki. The protocol was approved by the Ethics Committee for Clinical Trials (CESC) of the province of Vicenza and the ethics committee of San Bortolo Hospital (N.41/14). All participants were informed of the objectives of the study and signed for their informed consent.

### 2.2. Data Collection and Measurements

The data collected from patients were: age (years), gender, hypertension, diabetes mellitus, and causes of CKD classified by 4 categories (primary glomerular disease, genetic disease, secondary kidney disease, and unknown).

Blood samples in EDTA were collected during the outpatient visits in the PD patients or prior dialysis treatment after a long interdialytic interval in HD patients. The plasma was immediately centrifuged and stored at −80 °C.

We determined the urea, enzymatic creatinine, calcium, and phosphorus (P) (Dimension Vista^®^, SIEMENS Healthcare Diagnostics Inc., and Newark, Tarrytown, NY, USA). Parathyroid hormone (intact PTH) and 25 hydroxy vitamin D (25-(OH)D3) in serum were measured with the chemiluminescent immunoassay, CLIA (LIAISON^®^ XL instruments, DiaSorin, Saluggia (VC), Milano, Italy) as routine laboratory exams. 

The samples were thawed, and the serum FGF23 variable was measured simultaneously. 

The C-terminal FGF23 (cFGF23) levels were determined in plasma with a two-site enzyme-linked immunosorbent assay, ELISA (Immutopics, Inc., San Clemente, CA, USA). Two hundred microliters of plasma were used to assay the samples in duplicates. The samples with values greater than the highest standard were diluted by 1:10 or greater with the 0 pg/mL standard or optional sample diluent and reassayed. The result obtained was multiplied by the dilution factor. 

The residual urine output (UO) presence was collected from patients, and the quantity of UO (UO volume) was referred to as one day (mL/day).

### 2.3. Statistical Analysis

The continuous variables were summarized with the median and range (minimum-maximum) or the mean and standard deviation (±SD) as appropriate, and the categorical ones were summarized with the numbers and percentages of patients in each category.

All continuous variables were evaluated for normality with a Q-Q plot and with the Shapiro-Wilk test. The student’s *t*-test or the Mann-Whitney U test was applied to compare the continuous variables according to their normal or non-normal distribution. The categorical variables were compared with the chi-square or Fisher’s exact test. The correlations between the continuous variables were evaluated with Spearman’s rank correlation (R), and the 95% confidence interval (CI) was calculated using Fisher’s z transformation.

A univariable linear regression analysis was used to evaluate the relationship between the FGF23 natural log-transformed and potential predictors. All variables formed statistically significant results with a *p* < 0.05 and were considered in a multivariable linear regression model with a backward selection method at the 5% level of significance.

All reported *p*-values were two-sided, and the statistical significance was set at *p* < 0.05. The statistical analyses were performed with SAS 9.4 (SAS Institute Inc., Cary, NC, USA) for Windows.

## 3. Results

We enrolled 133 patients (median age, min-max: 66.0, 22.9–90.8 years): 58 HD patients (67.7, 33.6–90.8 years) and 75 PD patients (64.8, 22.9–87.6 years). The principal cause of CKD for all patients (also in the HD and PD groups) was secondary kidney disease (54.9%, HD: 51.7%, and PD: 57.3%). The residual UO was preserved in 93 (69.9%) patients. The baseline characteristics are summarized in Table 1 (full cohort, HD, and PD) and Table 2 (full cohort, patients without UO and with UO).

The cFGF23 levels were significantly higher in HD compared to PD patients (*p* = 0.0017) (as represented in Figure 1). Also, the levels of phosphorus (*p* = 0.0041) and 25-(OH) D3 (*p* < 0.0001) were significantly different.

We found a statistically significant negative correlation (R = −0.21; 95% CI: −0.37–0.03) between cFGF23 and the UO volume (*p* = 0.0250) (as represented in Figure 2), but the correlation was not significant when considering the type of dialysis treatment (PD: R= 0.05 95%CI: −0.20 0.30, *p* = 0.68 and HD: R = −0.08 95%CI: −0.33 0.19, *p* = 0.56) (Figure 3).

The residual urine output (UO) was preserved in 93 (69.9%) patients. The median cFGF23 level was higher in patients without UO (median, 1874.4 RU/mL; min-max 399.4–13,800.0) than in those with a preserved UO (median, 1400.1 RU/mL; min-max, 320.5–13,800) without reaching the statistical significance (*p* = 0.12).

Furthermore, when categorizing by residual urine output volume (UO volume < 100 mL/day; 100–499 mL/day; ≥500 mL/day), there was not a significant difference (*p* = 0.0983) (as represented in Figure 4)

The other laboratory variables that resulted in significant differences between patients without and with residual UO were creatinine (*p* = 0.0005), phosphorus (*p* = 0.0016), and 25-(OH) D3 (*p* = 0.0096).

To identify the factors associated with cFGF23 levels, we performed a linear regression analysis (Table 3). In the univariate analysis, the cFGF23 levels were positively correlated with creatinine (ß = 0.13441, *p* < 0.0001), urea (ß = 0.00704, *p* = 0.0233), and phosphorus (ß = 0.20957, *p* = 0.0010), and were inversely correlated with UO volume (ß = −0.00026, *p* = 0.0415). A significant association was also found with the type of dialysis (ß = 0.13441, *p* = 0.0415). The multivariable regression analysis showed that phosphorus (ß = 0.21677; *p* = 0.0007), type of dialysis (ß = −0.68392; *p* = 0.0003), and creatinine (ß = 0.08130; *p* = 0.0133) were significant and independent predictors of cFGF23 levels.

## 4. Discussion

Mineral and bone disorders are common in the CKD population, especially in the advanced stages; they represent a severe complication and an important risk factor, mortality-related, in patients requiring renal replacement therapy (extracorporeal or peritoneal) [1].

Hyperphosphatemia is frequently found in patients with CKD, as the progressive reduction in nephrogenic mass leads to a significant marked decrease in the ability of renal phosphorus excretion (which is taken with diet). FGF23 has a phosphaturic action, as it inhibits the reabsorption of phosphorus at the level of the proximal renal tubule. Hyperphosphatemia results in increased secretion of FGF23 from bone [9]. In advanced CKD, there is an exponential increase in FGF23, but it does not correspond to an adequate increase in klotho (its coreceptor), so at some point, FGF23 no longer works properly.

Some studies have shown that dialysis patients with residual UO have a better phosphorus balance than those without UO [17,18,19]. Concerning the phosphorus balance in PD and HD patients, the literature’s data are not unique, as some studies show better control in PD patients [17], even independently from residual UO [21], while other studies found higher levels of phosphorus in PD than in HD patients [22].

In our study, 133 patients were enrolled (median age, min-max: 66.0, 22.9–90.8 years), of which 58 patients had HD (67.7, 33.6–90.8 years) and 75 patients had PD (64.8, 22.9–87.6 years). The residual UO was preserved in 93 (69.9%) of 133 patients. Of the 75 patients with PD, 66 had UO (88%), and of the 58 patients with HD, 27 had UO (46.6%). In our cohort, the percentage of HD patients with a residual UO was higher than the percentage shown in other studies [24]. Nonetheless, our findings are not consistent with the literature, as patients with a residual UO had higher phosphorus levels than patients without UO. Concerning the phosphorus balance in HD and PD patients, our data showed higher phosphorus levels in PD than in HD patients. These contradictory results might be related to the fact that the phosphorus balance depends not only on the residual UO or the type of dialysis but also on the dialysis efficiency, dietary phosphorus burden, phosphate binders prescription, and patients’ compliance. Moreover, the phosphate balance was found to be also linked to other nutritional parameters, dialytic ages, and several medications other than phosphate binders [25]. However, the median values of phosphorus in HD and PD patients are, respectively, 4.5 and 5 mg/dL. In patients without and with residual UO, these values were 5 and 4.1 mg/dL. In all groups, the levels of phosphorus were within normal values.

The prevention of hyperphosphatemia in CKD patients using the dietary restriction of phosphate is mandatory. However, dialysis patients have a high risk of protein-energy wasting (PEW) due to multiple causes, such as inflammation, metabolic acidosis, endocrine disturbances, protein losses into the dialysate, and catabolism. Since proteins contain 15 mg of phosphate per gram, a compromise may be necessary, and at least 1.0–1.2 g protein/kg body weight/day is recommended for stable HD and PD patients [26]. Moreover, patients on PD lose 6 to 10 g of protein/day through their dialysate (higher than patients on HD), so a higher protein intake is recommended for these patients [27].

In a PD patient, despite an adequate intake of protein (>1 g/kg/day), when their residual renal function decreases, the need for protein decreases.

How could we explain why we found more controlled phosphorus values in patients without residual diuresis? Below we have formulated some hypotheses:

Some unconsidered aspects in the study, such as compliance with a diet low in phosphorus and adherence to chelation therapy, may have altered the obtained results; it could also be due to the fact that patients with preserved diuresis, precisely for this condition, felt freer (making a mistake) not to follow an adequate diet poor in phosphorus and therefore paradoxically this determined that they had higher median phosphorus values than the dialyzed patients without residual diuresis

In regards to FGF23, the analysis of Bi and colleagues [23] showed that FGF23 levels were significantly higher in HD patients than in PD patients. This data was also documented in our study and was an expected figure.

In our obtained data, the median cFGF23 was significantly higher in HD patients than in PD patients (2634.9 [340.8–13,800] vs. 1391.7 [320.5–13,800] UR/mL; *p* = 0.0017) (Table 1) and not significantly higher in patients without residual UO (1874.4 [399.4–13,800.0] vs. 1400.1 [320.5–13,800.0] RU/mL; *p* = 0,12) (Table 2).

The serum FGF23 values in HD patients are higher, probably because compared to the PD patient group, they had a lower percentage of residual UO, although they still had good control of phosphorus. This also explains why the FGF23 values did not change significantly despite the improved phosphorus control.

In our study, phosphorus (*p* = 0.0007) proved to be an independent predictor of serum FGF23 levels, and this could be explained by what was already published, i.e., that high values of phosphorus directly stimulate the secretion of FGF23 due to its phosphatic action.

The analyses carried out with our methods showed that the type of dialysis affected the levels of cFGF23. We found a significant negative correlation (R = −0.21; 95% CI: −0.37–0.03) between cFGF23 and the residual UO volume (*p* = 0.0250) (as represented in Figure 2), and in the univariate analysis, the cFGF23 levels inversely correlated with the UO volume (ß = −0.00026, *p* = 0.0415). However, the correlation was not significant when considering the type of dialysis (PD: R= 0.05 95%CI: −0.20 0.30, *p* = 0.68 and HD: R= −0.08 95%CI: −0.33 0.19, *p* = 0.56) (Figure 3) and in the multivariable regression analysis, as to say that in the relationship between the residual UO and FGF23, the type of dialysis represents a confounding factor; in fact, the type of dialysis is related to both diuresis and cFGF23.

As also described in the previous work on the re-classification of uremic toxins and their role in CKD [28], although all types of dialyzer remove small water-soluble compounds, the removal of large-middle molecules (>25–58 kDa) is definitely influenced by the type of dialyzer. Probably, the use of a non-new-generation dialyzer (the study took place between October 2014 and October 2015) influenced the value of c-FGF23 in HD patients. Therefore, it is reasonable to think that the decreased removal of FGF23 resulted in a greater accumulation of FGF23 in HD patients than in PD patients.

The principle behind the removal of urea toxins in PD is the diffusion of these from the blood through the peritoneal membrane into the peritoneal fluid, which is then eliminated after a certain period of time in the peritoneal cavity. The use of a highly permeable membrane together with better preservation of residual renal function determines that urea toxins are eliminated more efficiently through the peritoneal membrane and that the plasma and tissue levels of these molecules are lower in PD patients than in HD patients [29]. Moreover, PD is performed more continuously than HD, thus limiting the accumulation of uremic toxins such as FGF23.

### Limitations

(1)The cohort was obtained from a current study. Therefore, our studied population was not specifically chosen ad hoc; despite this, we defend the quality of the data obtained as they allowed us to confirm the data from the literature and provided us with points of reflection even for new investigations. What could be especially interesting is obtaining more data about the UO collection;(2)The two groups (patients in HD and patients in PD) were different in number and characteristics;(3)We only used single urinary output data and a single biohumoral examination data point;(4)The compliance with the phosphorus binding therapy and a low phosphorus diet was unknown.

## 5. Conclusions

In conclusion, cFGF23 was significantly higher in HD patients than in PD patients, and it was higher in patients without residual UO. We found a significant negative correlation between cFGF23 and the residual UO volume, but the correlation was not significant when considering the type of dialysis, as to say that in the relationship between the residual UO and FGF23, the type of dialysis represents a confounding factor. Our study found that the dialysis modality is an independent predictor of FGF23 levels. In particular, PD is associated with lower FGF23 levels than HD.

## Figures and Tables

**Figure 1 jcm-12-00222-f001:**
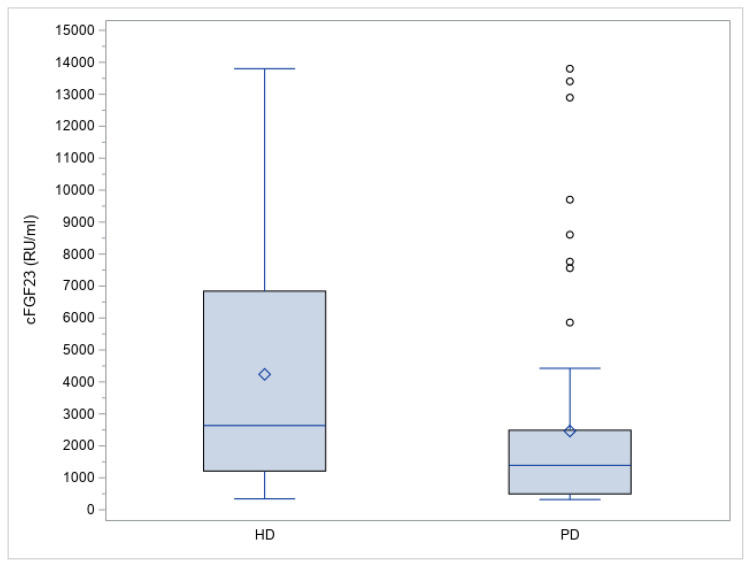
cFGF23 levels by type of renal replacement therapy (HD = hemodialysis, and PD = peritoneal dialysis).

**Figure 2 jcm-12-00222-f002:**
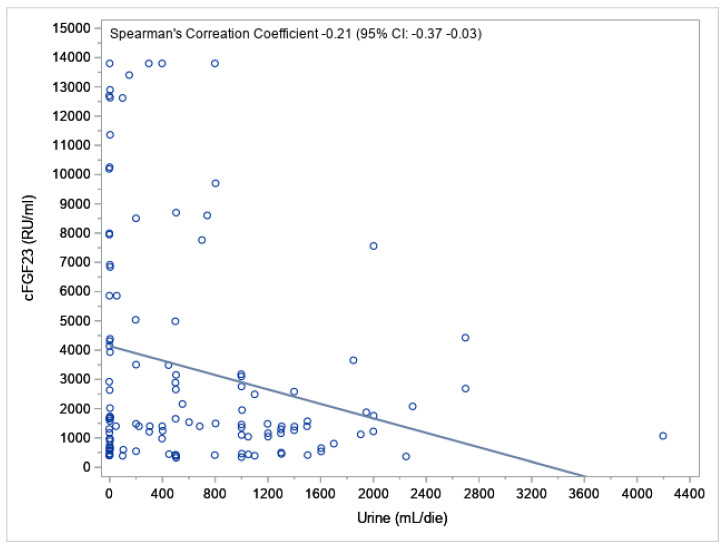
Scatter plot of the cFGF23 levels and residual urine output (UO) volume with the regression line and Spearman’s correlation coefficient.

**Figure 3 jcm-12-00222-f003:**
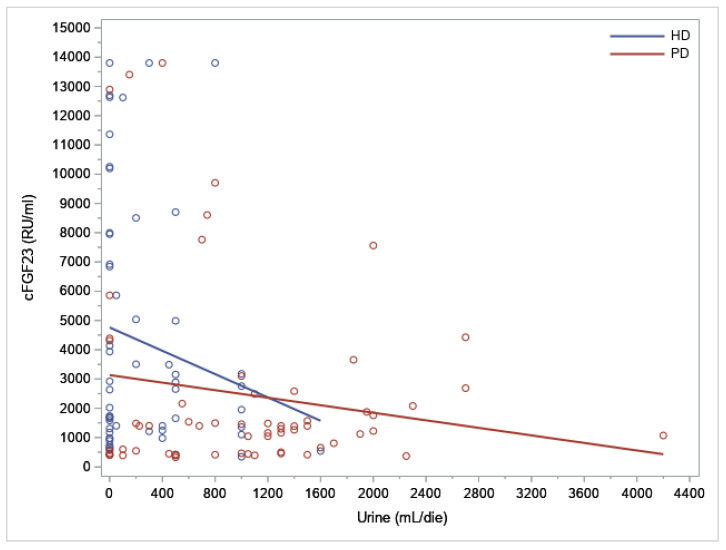
Scatter plot of the cFGF23 levels and residual urine output (UO) volume with the regression line considering the type of dialysis treatment (HD and PD).

**Figure 4 jcm-12-00222-f004:**
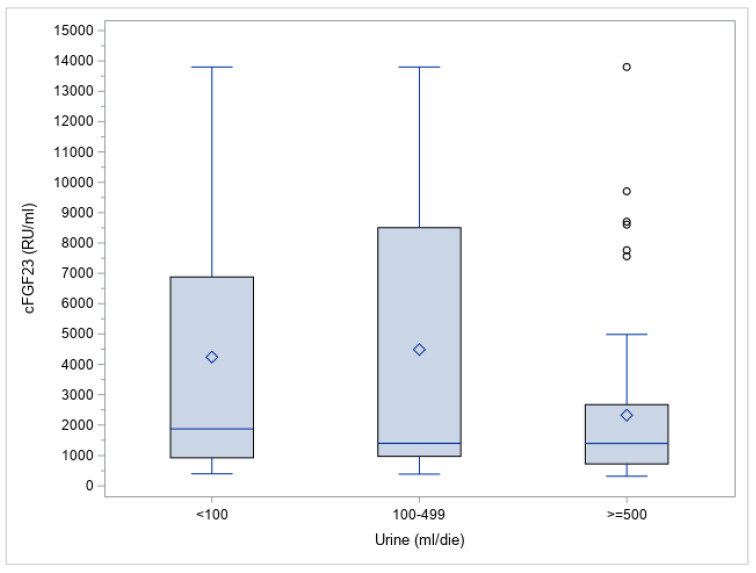
cFGF23 level distribution by residual urine output (UO) volume.

**Table 1 jcm-12-00222-t001:** Characteristics of the full cohort, i.e., the hemodialysis (HD) and peritoneal dialysis (PD) patients.

Variable	Full Cohort (N = 133)	HD (N = 58)	PD (N = 75)	*p* Value
Age (years)	66.0 (22.9–90.8)	67.7 (33.6–90.8)	64.8 (22.9–87.6)	0.46
Gender				1.00
Female sex (%)	39 (29.3%)	17 (29.3%)	22 (29.3%)	
Male sex (%)	94 (70.7%)	41 (70.7%)	53 (70.7%)	
Hypertension (%)	124 (93.2%)	54 (93.1%)	70 (93.3%)	1.00
Diabetes (%)	34 (25.6%)	16 (27.6%)	18 (24.0%)	0.64
Cause of CKD				0.90
Primary glomerular disease	24 (18.0%)	11 (19.0%)	13 (17.3%)	
Genetic disease	16 (12.0%)	7 (12.1%)	9 (12.0%)	
Secondary kidney disease	73 (54.9%)	30 (51.7%)	43 (57.3%)	
Unknown	20 (15.0%)	10 (17.2%)	10 (13.3%)	
Residual UO (%)	93 (69.9%)	27 (46.6%)	66 (88.0%)	<0.0001
Laboratory test				
Creatinine (mg/dL)	9.1 (3.2–19.5)	9.6 (3.8–18.6)	8.8 (3.2–19.5)	0.11
Urea (mg/dL)	129.3 ± 33.3	136.7 ± 36.1	123.6 ± 30.1	0.024
Calcium (mg/dL)	9.0 (7.6–12.8)	9.0 (7.9–10.4)	9.2 (7.6–12.8)	0.11
Phosphorus (mg/dL)	4.8 (2.1–8.9)	4.5 (2.1–8.9)	5.0 (2.5–8.9)	0.0041
iPTH (pg/mL)	99.0 (4.0–1010.0)	167.0 (4.0–1010.0)	80.0 (7.0–732.0)	0.065
25-(OH) D3 (ng/mL)	16.8 (5.0–71.9)	21.7 (5.0–71.9)	11.5 (5.0–29.8)	<0.0001
cFGF23 (RU/mL)	1493.1 (320.5–13,800.0)	2634.9 (340.8–13,800.0)	1391.7 (320.5–13,800.0)	0.0017
UO volume (mL/day)	500.0 (0.0–4200.0)	0.0 (0.0–1600.0)	1000.0 (0.0–4200.0)	<0.0001

**Table 2 jcm-12-00222-t002:** Characteristics of the full cohort, without the residual urine output (without UO) and with the residual urine output (with UO) patients.

Variable	Full Cohort (N = 133)	Without UO (N = 40)	With UO (N = 93)	*p* Value
Age (years)	66.0 (22.9–90.8)	58.5 (33.6–86.7)	68.0 (22.9–90.8)	0.093
Gender				0.029
Female sex (%)	39 (29.3%)	17 (42.5%)	22 (23.7%)	
Male sex (%)	94 (70.7%)	23 (57.5%)	71 (76.3%)	
Hypertension (%)	124 (93.2%)	38 (95%)	86 (92.5%)	0.72
Diabetes (%)	34 (25.6%)	16 (27.6%)	18 (24.0%)	0.64
Cause of CKD				0.0016
Primary glomerular disease	24 (18.0%)	14 (35.0%)	10 (10.8%)	
Genetic disease	16 (12.0%)	5 (12.5%)	11 (11.8%)	
Secondary kidney disease	73 (54.9%)	13 (32.5%)	60 (64.5%)	
Unknown	20 (15.0%)	8 (20%)	12 (12.9%)	
Laboratory test				
Creatinine (mg/dL)	9.1 (3.2–19.5)	10.3 (5.3–18.6)	8.7 (3.2–19.5)	0.0005
Urea (mg/dL)	129.3 ± 33.3	129.7 ± 37.5	129.1 ± 31.6	0.93
Calcium (mg/dL)	9.0 (7.6–12.8)	9.0 (8.1–10.4)	9.1 (7.6–12.8)	0.87
Phosphorus (mg/dL)	4.8 (2.1–8.9)	4.1 (2.1–8.9)	5.0 (2.1–8.9)	0.0016
iPTH (pg/mL)	99.0 (4.0–1010.0)	181.0 (4.0–1010.0)	82.5 (7.0–732.0)	0.20
25-(OH) D3 (ng/mL)	16.8 (5.0–71.9)	20.8 (5.0–71.9)	16.6 (5.0–54.9)	0.0096
cFGF23 (RU/mL)	1493.1 (320.5–13,800.0)	1874.4 (399.4–13,800.0)	1400.1 (320.5–13,800.0)	0.12

**Table 3 jcm-12-00222-t003:** Univariable and multivariable regression analysis of the factors associated with cFGF23 levels (RU/mL); abbreviations: ß, the regression coefficient of the univariable or multivariable linear regression; CI, confidence interval.

Variable	Univariable			Multivariable		
	ß	95% CI	*p* Value	ß	95% CI	*p* Value
Age	−0.02062	−0.03517; −0.00607	0.0059			
Sex (F)	−0.29946	−0.73660; 0.13768	0.1775			
Dialysis (PD)	−0.62306	−1.00102; −0.24509	0.0014	−0.68392	−1.04698; −0.32086	0.0003
Residual UO	−0.35999	−0.77786; 0.05789	0.0906			
Hypertension	0.76648	−0.12389; 1.65686	0.0909			
Diabetes	0.28007	−0.16636; 0.72649	0.2166			
Urine (mL/day)	−0.00026	−0.00052233; −0.00001042	0.0415			
Creatinine (mg/dL)	0.13441	0.07054; 0.19828	<0.0001	0.08130	0.01723; 0.14537	0.0133
Urea (mg/dL)	0.00704	0.00097556; 0.01311	0.0233			
Calcium (mg/dL)	−0.11283	−0.40967; 0.18402	0.4531			
Phosphorus (mg/dL)	0.20957	0.08667; 0.33247	0.0010	0.21677	0.09372; 0.33982	0.0007
iPTH (pg/mL)	0.00092	−0.00034267; 0.00219	0.1514			
25-(OH) D3 (ng/mL)	0.00812	−0.01006; 0.02630	0.3784			

## Data Availability

Not applicable.

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
