# Peer review of "Relationship between Residual Urine Output and Type of Dialysis with FGF23 Levels"

_jcm, 2022, doi:10.3390/jcm12010222_

Round 1
Reviewer 1 Report (Previous Reviewer 3)
This study seems to be promising and very interesting, however some points need to be clarified, otherwise the paper is almost ready for publication.
1. FGF-23 levels were measured in blood samples collected during outpatient visits in PD patients or prior dialysis treatment after long interdialytic interval in HD patients. However, duration of treatment of PD or HD may play a major role for FGF-23 levels. The study will be more completed if the writers had FGF-23 levels measured immediate after dialysis. Can you place a comment about that?
2. In addition, it is very important to know how many years were the patients in a dialysis method and if anyone had dropped out for a method to another due to dialysis problems. Add this information.
3. A limitation of the study, although was mentioned by the writers, is the compliance with phosphorus binding therapy. The P levels between the two groups showed significant difference so that can be problematic for the results of FGF-23 levels. I suggest the authors to try to make a correlation between the two groups after changes of the sample (to exclude certain patients from the study in order to normalize the sample), so the P levels be about the same.
4. Some points in the text after a grammatical and linguistic recheck have to be rewritten.
Author Response
Response to Reviewer 1 Comments
This study seems to be promising and very interesting, however some points need to be clarified, otherwise the paper is almost ready for publication
Point 1: FGF-23 levels were measured in blood samples collected during outpatient visits in PD patients or prior dialysis treatment after long interdialytic interval in HD patients. However, duration of treatment of PD or HD may play a major role for FGF-23 levels. The study will be more completed if the writers had FGF-23 levels measured immediate after dialysis. Can you place a comment about that?
Response 1: Thanks for the suggestion but we have not collected this data. Blood samples in EDTA were collected during outpatient visits in PD patients or before dialysis treatment after a long interdialytic interval in HD patients. This work is part of a larger study in which we were interested to follow-up FGF23 and other parameters over time (but not before and after dialysis).
Point 2: In addition, it is very important to know how many years were the patients in a dialysis method and if anyone had dropped out for a method to another due to dialysis problems.
Response 2: Thanks for the suggestion but we have not collected this data. Probably there were some missing data and for this reason we could not enter it.
Point 3: A limitation of the study, although was mentioned by the writers, is the compliance with phosphorus binding therapy. The P levels between the two groups showed significant difference so that can be problematic for the results of FGF-23 levels. I suggest the authors to try to make a correlation between the two groups after changes of the sample (to exclude certain patients from the study in order to normalize the sample), so the P levels be about the same.
Response 3: Thanks for the suggestion Unfortunately we are unable to execute this request. We entered and analyzed all the data we had available.
Point 4: Some points in the text after a grammatical and linguistic recheck have to be rewritten.
Response 4: The manuscript has been checked by a native English-speaking colleague.

Reviewer 2 Report (Previous Reviewer 2)
Please see attached.

Author Response
Response to Reviewer 2 Comments
The manuscript performed a cross-sectional study in hemodialysis (HD) and peritoneal dialysis (PD) patients, to determine if the dialysis modality and the UO influence FGF23 levels. The article is well written and composed. There are some points of revised version need to be improved.
Point 1: On page 6, Figure 3, title is missing.
Response 1: Figure 3: Scatter plot of cFGF23 levels and residual urine output (UO) volume with regression line considering type of dialysis treatment (HD and PD)
Point 2: On page 8, line 191, “p=0.0096” is not consistent with other text fonts.
Response 2: p=0.0096

This manuscript is a resubmission of an earlier submission. The following is a list of the peer review reports and author responses from that submission.
Round 1
Reviewer 1 Report
The study presents FGF23 differences in HD vs. PD as well as in anuric vs. non-anuric patients, also correlation between FGF23 and urine output. The topic is important as mineral and bone disorders are frequent in CKD. Below my comments:
1. In Abstract we read „ statistical significance was set at p<0.05” and then “ The median cFGF23 were significantly (...) higher in patients without UO (p=0.12)". 0.12 is higher than 0.05 hence the conclusion is not correct.
2. Sentence (page 2-3): „Samples with values greater than the highest standard were diluted 1:10 or greater with the 0 references 98 units [RU] per milliliter Standard or optional Sample Diluent reagent and re-assayed.” is not easy to read, please re-write or at least please remove capital letters.
3. Please try to put regression line for all points from Figure 2 in Figure 3. Now, the only difference between Figures 2 and 3 is regression line for all points.
4. Please check the correlation between FGF23 and Urine only in non-anuric patients. Is it correct to investigate the relationship between cFGF23 and Urine in patients without urine (?).
5. It is not clear if there are median and range or mean and SD in all tables.
6. Please make bold p-value for urea in Table 1.
7. Please make bold p-value for Female sex in Table 2.
8. In Results we read “The principal cause of CKD for all patients (also in HD and PD groups) was secondary kidney disease (54.9%, HD: 123 51.7% and PD: 57.3%).” What is “secondary kidney disease”?
9. Using gathered material it is possible to show also correlations between phosphorus, PTH, vit. D and FGF23, please consider.
10. Please add p-value to the figures, then figures are much more readable.
11. Please check again correlation between FGF23 and Urine volume in HD (Figure 3).
12. Should be day instead of “die” as unit of urine output (?).
13. Throughout text we have three repetitions of the same information: “Residual UO was preserved in 93 (69.9%) patients.”, “Residual urine output (UO) was preserved in 93 (69.9%) patients.”, “Residual UO was preserved in 93 (69.9%) of 133 patients.”
14. Please check numbers in the sentence “A significant association was also found with type of dialysis (ß = 0.13441, p=0.0415)” as they are not consistent with numbers in Table 3.
15. Is it necessary to use 5 digits after comma in Table 3 and the same in text (?).
16. In Results we read: “The median cFGF23 156 level was higher in patients without UO (median, 1874.4 RU/ml; min-max 399.4-13800.0) than in those who preserved UO (median, 1400.1 RU/ml; min-max, 320.5-13800) without reaching the statistical significance (p=0.12).” and in the Discussion: “In the data that we obtained, the median cFGF23 was (...) higher in patients without residual UO (1874,4 [399,4-13800,0] vs 1400,1 [320,5 -13800,0] RU/mL; p=0,12) (Tab.2).”. Please do not repeat the same numbers in Discussion as we have them in Results and in table. We cannot say that median was higher if p value is higher than 0.05.
17. What is the relationship between this paragraph: “In recent years, dialyzer clearance profiles and membranes have improved considerably. These include new permeability indices, hydrophilic or hydrophobic membrane nature, adsorption capacity, and electrical potential.” with the material discussed in the manuscript.
18. Why “Data Availability Statement” is “Not applicable” in this study?
Reviewer 2 Report
Comments to the Author
The manuscript performed a cross-sectional study in hemodialysis (HD) and peritoneal dialysis (PD) patients, to determine if the dialysis modality and the UO influence FGF23 levels. The article is well written and composed. There are some points need to be improved.
1. On page 2, line 74, the patient admission and discharge criteria can be described in more detail in Study Population.
2. On page 2, line 82, contents related to methodology validation should be reported in Data collection and measurements.
3. On page 3, Table 1, the gender variable in Table 1 only shows the number of female, but also the number of male.
4. On page 5, Figure 1 and Figure 2, the annotation of the figures should not be placed on the left side of the figure, but should be placed below the figure.
5. On page 6, Figure 3, legend for empty circle is missing.
6. On page 8, line 179, “Mineral and bone disorders are common in the CKD population, especially in advanced stages;they represent a severe complication and an important risk factor, mortality-related, in patients requiring renal replacement therapy (either extracorporeal or peritoneal).” needs reference.
7. On page 9, line 271, more limitations of methodology and how to cause bias can be added to limitations.
Reviewer 3 Report
This study seems to be promising and very interesting, however some points need to be clarified, otherwise the paper is almost ready for publication
FGF-23 levels were measured in blood samples collected during outpatient visits in PD patients or prior dialysis treatment after long interdialytic interval in HD patients. However, duration of treatment of PD or HD may play a major role for FGF-23 levels. The study will be more completed if the writers had FGF-23 levels measured immediate after dialysis. Can you place a comment about that?
In addition, it is very important to know how many years were the patients in a dialysis method and if anyone had dropped out for a method to another due to dialysis problems.
A limitation of the study, although was mentioned by the writers, is the compliance with phosphorus binding therapy. The P levels between the two groups showed significant difference so that can be problematic for the results of FGF-23 levels. I suggest the authors to try to make a correlation between the two groups after changes of the sample (to exclude certain patients from the study in order to normalize the sample), so the P levels be about the same.Some points in the text after a grammatical and linguistic recheck have to be rewritten.